# In Vitro Modeling of Non-Solid Tumors: How Far Can Tissue Engineering Go?

**DOI:** 10.3390/ijms21165747

**Published:** 2020-08-11

**Authors:** Sandra Clara-Trujillo, Gloria Gallego Ferrer, José Luis Gómez Ribelles

**Affiliations:** 1Center for Biomaterials and Tissue Engineering (CBIT), Universitat Politècnica de València, 46022 Valencia, Spain; ggallego@ter.upv.es (G.G.F.); jlgomez@ter.upv.es (J.L.G.R.); 2Biomedical Research Networking Center on Bioengineering, Biomaterials and Nanomedicine (CIBER-BBN), 46022 Valencia, Spain

**Keywords:** blood cancer, disease modeling, bone marrow, niche, microenvironment, tissue engineering, 3D models, tumor-on-a-chip, leukemia, myeloma

## Abstract

In hematological malignancies, leukemias or myelomas, malignant cells present bone marrow (BM) homing, in which the niche contributes to tumor development and drug resistance. BM architecture, cellular and molecular composition and interactions define differential microenvironments that govern cell fate under physiological and pathological conditions and serve as a reference for the native biological landscape to be replicated in engineered platforms attempting to reproduce blood cancer behavior. This review summarizes the different models used to efficiently reproduce certain aspects of BM in vitro; however, they still lack the complexity of this tissue, which is relevant for fundamental aspects such as drug resistance development in multiple myeloma. Extracellular matrix composition, material topography, vascularization, cellular composition or stemness vs. differentiation balance are discussed as variables that could be rationally defined in tissue engineering approaches for achieving more relevant in vitro models. Fully humanized platforms closely resembling natural interactions still remain challenging and the question of to what extent accurate tissue complexity reproduction is essential to reliably predict drug responses is controversial. However, the contributions of these approaches to the fundamental knowledge of non-solid tumor biology, its regulation by niches, and the advance of personalized medicine are unquestionable.

## 1. Introduction

Historically, the objective of tissue engineering and regenerative medicine (TERM) has been to apply the principles of engineering and life sciences to the development of biological substitutes that restore, maintain, or improve the function of a tissue or whole organ [1]. While this objective remains intact, the focus in the field has been extended to the implementation of engineered tissues that will never be implanted into patients, but will transform the way we study human tissue physiology in vitro [2,3,4,5]. Each tissue and organ is unique and has well defined functions, anatomy and cellular, molecular and soluble components. In vivo, individual cells are harbored in specific niches where they integrate many external cues (including those that arise from extracellular matrix (ECM), mechanical stimulation and soluble signals from adjacent and distant cells) to generate a basal phenotype and respond to perturbations in their environment. The development of 3D platforms with well-defined architectures resembling native cellular environments has contributed to significant advances, among other tissues, in liver or heart modeling [6,7,8]. The integration of three dimensionality, multi-cellular interactions, patient-specific polymorphisms, fine control of chemical parameters (pH, oxygen level, biochemical gradients) and ECM composition are the main assets of this engineered tissues [4,9,10].

## 2. Modeling Solid Tumors in Vitro

Cancer is a heterogeneous dynamic disease in which the associated stroma plays a critical role as a pro-tumorigenic environment, drug desensitization inductor and drug penetration barrier [11]. 3D engineered cancer models have been used to overcome major issues of conventional 2D planar cultures and animal models. The average success rate for candidate drugs in translating from animal models to clinical cancer trials is less than 8% [12]. Biological differences among humans and animals limit their ability to mimic complex processes such as carcinogenesis and tumor physiology, progression and metastasis. Mice are the most frequently used animal models. Crucial genetic, molecular, immunologic and cellular differences between mice and humans prevent them from serving as effective models [9]. Significant progress has been made, such as humanizing mice by transplanting human cells or obtaining patient-derived tumor xenografts (so called PDTX or avatar mouse). Nevertheless, such models are still challenging and expensive to adopt for routine use. Furthermore, fundamental differences in telomerase regulation between rodents and humans [13] have raised questions regarding the reliability of transgenic and inducible mouse cancer models, and discrepancies between certain rodent and human cytokines generate uncertainty for mouse models [9,14]. 2D planar cultures’ lack of architecture, cell-cell and cell-ECM interactions, and the exposure of cells to high-stiffness substrates like culture plates could affect cell behavior in terms of gene expression profile and drug sensitivity. For example, the PI3K–AKT–mTOR pathway is a central regulator of cell growth, proliferation, survival, metabolism and aging. Riedl et al. reported significant differences in mTOR activity and crosstalk between AKT-mTOR-S6K and the MAPK pathway in spheroids vs. planar cultures of colorectal cancer Caco-2 cells, including alterations in the responses in treatments with inhibitors of AKT, mTOR and S6K axis or of the MAPK (ERK) axis, which are ongoing pharmacological targets [15]. Moreover, the role of specific ECM signaling in regulating gene expression and cell fate has been largely validated as a pivotal agent in cancer progression and drug resistance. The attachment of tumor cells to the ECM may trigger cell adhesion-mediated drug resistance (CAM-DR). Several receptors such as integrins and their ligands, including fibronectin (FN) or hyaluronic acid (HA), are involved in this process. The interaction between α4β1 integrin on tumor cells and FN induces progressive drug resistance in chronic lymphocytic leukemia (CLL) and acute myeloid leukemia (AML). Also, β1 integrin-mediated PI3K activation overrides treatment-induced cell cycle arrest and apoptosis in various solid tumors [16]. As the specific crosstalk between a given cancer and its stroma varies for each cancer type and perhaps for each patient, in vitro models that better reflect the in vivo human environments and their heterogeneity may provide more accurate indications of patient outcome [17,18].

TERM has been used to explore several approaches for modeling solid tumors (Table 1). Scaffold-free models such as spheroids and organoids have achieved great in vitro results [15,19]. Organoids, cell aggregates deriving from one or several stem cells able to self-organize and phenocopy essential aspects of the organs they derive from, are of great interest from the point of view of drug testing, as they are easily compatible with high throughput screening technologies (HTS). Genetic modification of organoids allows disease modeling and organoids can be grown from patient tumor tissues (tumoroids) and recapitulate better native tumors arising in superior models for patient-specific drug testing [20]. Other approaches incorporate polymeric substrates with tunable composition, stiffness or functionality into the equation, as in scaffold or hydrogel-based models [21,22]. Including bioreactors and perfused microfluidic chambers gives strict control of oxygen, temperature, pH or nutrients and precise spatiotemporal control over gradient formation [23]. Advances in bioprinting techniques endow tumor-on-a-chip models with specific properties such as anisotropy or complex physiological architecture [22,23]. More recent approaches integrate the above-mentioned features in realistic systems that can even include cancerous vascularized tissues embedded in chemico-physically defined environments with ECM and healthy neighboring cells under dynamic perfused conditions.

## 3. Blood Cancers

Blood cancers, non-solid tumors or hematological malignancies are a collective term for neoplastic diseases of the hematopoietic and lymphoid tissues with a clinical presentation as leukemia, lymphoma or myeloma [25]. In humans, definitive adult hematopoiesis is established in the bone marrow (BM) and thymus [26]. Through this process, all blood cells arise from the hematopoietic stem cell (HSC). Precursor cells proceed through specific maturation steps before leaving the BM as mature circulating cells. Two major lineages exist: thrombocytes, erythrocytes, granulocytes, and dendritic cells are derived from the common myeloid progenitor (CMP) and make up the myeloid lineage, while T- and B-lymphocytes, plasma cells and natural killer cells arise from the common lymphoid progenitor (CLP) and compose the lymphoid lineage [26,27]. Lineage and maturation stages can be assessed by morphologic, immunophenotypic, genetic and cytochemical features. Hematological malignancies are categorized by the same methods according to cell origin, maturation or tumor characteristics [25]. The World Health Organization (WHO) has published a unified classification of neoplastic diseases of the hematopoietic and lymphoid tissues [28,29]. Two major classes of cell types are primarily affected: myeloid neoplasms or lymphoid neoplasms. And then, rare histiocytic, dendritic or mast cell neoplasms. Each of these major groups is categorized in different subclasses, the most common of which are summarized in Figure 1.

Although leukemias, lymphomas, and myelomas share some common features, they also have major differences, with similarities and differences in each disease group. Leukemia (a term derived from the Greek words “leukos” and “heima”) refers to an excess of leukocytes in the body and originates in the BM. It can arise in either of two main groups -lymphocytes or myelocytes- and can be acute (a rapidly progressing form in which affected cells are very immature and unable to accomplish their function) or chronic (which progresses slowly from cells that are relatively differentiated but crudely functional). Lymphoma involves lymphocytes and is initiated in lymphoid tissues. Non-Hodgkin’s lymphoma is the most prevalent form, with more indolent forms that progress slowly with well differentiated cells, and the more aggressive forms with less differentiated lymphocytes. Hodgkin’s disease has different clinical features and is characterized by the presence of the distinctive Reed-Stenberg Cells. Myeloma are plasma cell disorders characterized by clonal proliferation of malignant cells, normally in the BM [30]. In leukemia, the cancerous cells are discovered circulating in the blood and BM, while in lymphoma cells tend to aggregate and form tumors in lymphatic tissues. Myeloma is mainly a tumor of the BM (Box 1). This review will focus on hematological malignancies with BM homing, how they are supported by BM niches, and explains how tissue engineers could take advantage of these interactions and architectures to recapitulate the malignant process in vitro.

Box 1Blood cancers.“In leukemia, the cancerous cells are discovered circulating in the blood and BM, while in lymphoma cells tend to aggregate and form tumors in lymphatic tissues. Myeloma is mainly a tumor of the BM.”

## 4. Bone Marrow Microenvironment, Home of Hematological Malignancies

### 4.1. Healthy BM Niche: An Intricate and Precisely Organized Network Sustaining HSCs Homeostasis

To ensure effective hematopoiesis throughout an individual’s lifespan, HSCs are tightly regulated in their complex BM niche (BMN), which keeps a balance between maintaining the HSC population and producing mature immune and blood cells. Stromal cells, ECM and biochemical gradients orchestrate the regulation between quiescence and self-renewal that contribute to maintaining the HSCs vs. their activation, lineage commitment and terminal differentiation. The hematopoietic system has developed an adaptive response capacity in order to maintain homeostatic replacement of blood and immune cells (steady-state hematopoiesis), and also to rapidly increase differentiated cell production in a context of acute blood loss, infection and metabolic or toxic stress (emergency hematopoiesis). Aberrant prioritization of differentiation over HSC self-renewal and quiescence leads to the exhaustion of the HSC compartment, while inhibition of differentiation involves ineffective blood production. In both cases, the hematopoietic system becomes exhausted, leading ultimately to BM failure and hematological malignancies. Under physiological conditions, functional HSC heterogeneity appears to be controlled by spatially-different niches. HSCs with diverse differentiation biases (myeloid vs. lymphoid) or different fates (quiescent vs. activated) have been found to lodge in different anatomical locations under the control of different stimuli [31]. Despite the controversies, it is generally accepted that there are two main HSC niches in the BM: the endosteal BMN (EBMN) in the endosteum, with low vascularity (arteries and arterioles) and the central BMN (CBMN), with higher vasculature, arterioles and sinusoids, and enriched in HSCs. In both niches HSCs reside mainly in perivascular areas where endothelial cells, different stromal cells of mesenchymal origin, neurons or Schwann cells and mature blood or immune cells critically regulate their location, number and fate (Box 2). They are also governed by other factors such as the abundance of oxygen tension, shear flow or reactive oxygen species (ROS) (Figure 2).

The central niche comprises > 90% of the BM volume, and shelters 85% of the HSCs. It contains most sinusoids and arterioles. Specific and differentiated functions are linked to arteriolar (aBMN) or sinusoidal (sBMN) BM niches. Sinusoidal areas are tightly related with myelopoiesis and contain CMPs and steady-state hematopoiesis. In sBMN mesenchymal stem cell (MSCs) subpopulations such as leptin receptor expressing cells and abundant reticular cells are mainly overlapping (LepR^+^ CAR MSCs) and produce crucial growth factors for HSC maintenance as CXC motif chemokine ligand 12 (CXCL-12) and stem cell factor (SCF). In arteriolar perivascular areas of CBMN and EBMN, nestin and neural glial antigen expressing cells (Nest^+^ NG2^+^ MSCs) are the most abundant MSCs subpopulation and also produce soluble factors involved in HSC maintenance, such as CXCL-12 and SCF. Sinusoidal (sEC) and arteriolar (aEC) endothelial cells have also been reported to provide soluble factors related to HSC maintenance such as CXCL-12, SCF, angiopoietin (ANG-1) or fibroblast growth factor 2 (FGF-2). However, they have phenotypical differences that determine different functions; aECs form less permeable blood vessels. High permeability of adjacent vessels increases ROS intracellular levels in HSCs, increasing their migration capacity while compromising their self-renewal, hampering quiescence and accelerating differentiation. Sinusoids therefore promote HSC activation and lodge immature and mature leukocyte trafficking to and from the BM, while more quiescent HSCs are found adjacent to arterioles. The ROS^low^ HSCs found in sinusoidal areas reside near megakaryocytes, which in the BM niche promote HSC quiescence through secretion of transforming growth factor beta (TGF-β), thrombopoietin (TPO) or platelet factor 4 (PF-4). sECs also express higher levels of E-selectin (adhesion molecule that supports HSC activation) than aECs [32,33]. Sympathetic nerve fibers innervate BM perivascular areas and contribute to their functional differentiation. Non myelinating Schwann cells (nmSCs) in sBMN regulate migration of HSCs by direct contact [31], while Nest^+^ nmSCs maintain hibernating HSCs in aBMN [34]. nmSCs associated with sympathetic nerve fibers can promote HSC quiescence by activating latent TFG-β in EBMN and CMBN [33]. Sympathetic nerves release soluble mediators such as catecholamines from terminals. Noradrenaline reduces CXCL-12 production by different niche-forming cells [35], inducing HSC activation (Figure 2c).

The EBMN is a much smaller niche (less than 10% of total BM volume) with 15% of the total HSC population. Several EBMN signals are related to the promotion of HSC quiescence, essential to preserve hematopoiesis normalcy through lifespan. It is related to lymphopoiesis and harbors CLPs [31]. Regulation of hematopoiesis by osteoblasts (OB) may depend on their differentiation state [36]. The contribution of the osteoblastic niche to HSC maintenance remains controversial due to OB population heterogeneity and varying degree of maturation. OBs produce cytokines and growth factors that promote HSC self-renewal and quiescence. For example, ANG-1 interacts with ANG-1 receptor TIE-2 on HSCs to promote quiescence and adhesion. Increased expression of Jagged-1 (JAG-1) in OBs simultaneously increases their number and HSC self-renewal through Notch-1 signaling activation in HSCs [36]. OBs also appear to regulate HSC homing and engraftment in the EBMN after HSC transplantation. Osteocytes, mature bone cells entrapped in the calcified bone matrix, control HSCs through the secretion of the granulocyte-colony stimulating factor (G-CSF). Osteoclasts (OCs), bone degrading cells of monocytic origin, also affect hematopoiesis [36].

Non-cellular elements such as oxygen level also affect HSCs in different niches. The hematopoietic compartment is relatively hypoxic, which contributes to HSC pluripotency by mechanisms such as reducing intracellular ROS. The oxygen level close to arterioles is higher than in perisinusoidal regions [32,35]. The ECM produced by niche cells provides structural integrity and has a regulatory effect on niche-forming cells and HSCs [18]. The most abundant proteins are FN, collagens (COL) from I to XI, tenascin, osteopontin, thrombospondin or elastin. Also important are proteoglycans, which can present glycosaminoglycan side-chains such as HA, chondroitin sulfate, heparan sulphate or heparin. ECM interactions with cells are mediated by integrins or selectins and membrane-bound immunoglobulins such as the intercellular adhesion molecule 1 (ICAM-1) or vascular cell adhesion molecule 1 (VCAM-1) [32,37]. Proteoglycans and collagens have been shown to be essential for HSC maintenance; for example HA is required for in vitro hematopoiesis. As in cell-cell interaction, cell-ECM interactions are differentially regulated during steady-state or emergency hematopoiesis [37]. The mechanical properties of BM determined by ECM composition have also been shown to affect HSC fate. HSC cultivation in different substrates of varying stiffness and topographies have shown that matrix density and biophysical properties, e.g., specific presentation of adhesion ligands, contribute to HSC niche modulation, either directly or indirectly. ECM density and components also modulate direct cell-cell communications by conditioning secreted factors [37].

Box 2Bone marrow microenvironment.Healthy BMN in a nutshell. “HSCs maintenance vs. differentiation is tightly regulated in their complex BM niche (…). HSCs reside mainly in perivascular areas, where ECs, different stromal cells of mesenchymal origin, neurons or Schwann cells and mature blood or immune cells and factors as oxygen tension, shear flow or ROS regulate their location and fate (…). The central BMN is related with myelopoiesis and steady-state hematopoiesis; sinusoidal areas promote HSCs activation and lodge immature and mature leukocyte trafficking while more quiescent HSCs are found adjacent to arterioles. In bone proximities, the endosteal BMN is related with promotion of HSCs quiescence”.

### 4.2. BM Niche in Hematological Malignancies: When the Regulatory Machinery Becomes the Perfect Tumor Partner

Multiple intricate interactions maintain hematopoiesis physiology and also the pathophysiology of hematological malignancies, which alter the BMN and its normal interactions and so contribute to tumor progression. Although BMN was suggested to influence tumor initiation twenty years ago its role is still not clear. However, there is significant evidence of BMN and malignant cell crosstalk being involved in tumor progression and resistance to therapies [31]. Transformed cells compete with HSCs to occupy the BMN during the progression of the disease, which disrupts physiological interactions, reduces the HSC population and disturbs normal hematopoiesis. For example, malignant cells have a dependence on canonical HSC niche pathways, such as CXCL12- C-X-C chemokine receptor type 4 (CXCR4). However, as the disease progresses transformed cells progressively become independent of BMN control.

Niche-driven transformations are mutations or functional alterations in BMN-forming cells that predispose to myeloid malignant tumors. The first indication was provided by two studies reporting that genetic ablation of the tumor suppressor gene retinoblastoma (Rb) [38] or retinoic acid receptor γ (Rar_γ_) [39] in mice induced myeloproliferative neoplasms (MPN), even though their development required inactivation of either of these genes in hematopoietic and BMN-forming cells. Development of MPN from Rar_γ_ mutations needs increased levels of the tumor necrosis factor (TNF) [39], depicting a pro-inflammatory environment as a crucial triggering factor, together with BMN alteration. As inflammation is a hallmark of aging, and myeloid malignancies are more prevalent among the elderly, inflamed BMN could be used as a tumor-initiating factor. As BMN aging promotes myeloid biases at the expense of lymphoid differentiation, some authors suggest that it facilitates pre-malignant clone growth by overstimulating myeloid cell expansion, which can lead to myeloid malignancies. This means that myeloid malignancies may develop over a period of years as a continuous process involving simultaneous mutation of pre-malignant cells caused by certain BMN alterations [40,41,42]. There is further evidence that BMN is a predisposing factor from the fact that many recipients contract leukemia after human clinical halogenic HSC transplantations as a result of the transformation of healthy HSCs into malignant clones [43].

BMN remodeling by malignant cells contributes to disease progression. Malignant cells alter the transcriptome, proteome and function of BMN-forming cells by means or secreted factors, exosomes or direct cell-cell contact and promote BMN changes towards angiogenesis and inflammation [44,45,46]. MSCs, adipocytes, OBs, ECs and sympathetic neurons or Schwann cells are also affected. The reprogramming of BMN-forming cells has been described in myeloid and lymphoid malignancies. In the case of myeloid malignancies, chronic myeloid leukemia (CML) cells activate MSCs through soluble factors like CC motif ligand 3 (CCL3) or TPO, and by direct cell-cell contact causing overproduction of functionally altered OBs that do not support normal HSC maintenance [47]. In lymphoid malignancies, T cell acute lymphoblastic leukemia (T-ALL) cells impair normal hematopoiesis with a dramatic loss of OBs [48]. In CLL, exosomes from neoplastic clones reduce the production of soluble HSC supporting factors, and CLL cell-derived vesicles affect immune cell function as natural killer cells and accelerate the transition of stromal cells towards cancer supportive phenotypes, up-regulating production of angiogenic factors by ECs and MSCs [49]. Sympathetic neurons, which innervate BMN, also become altered by malignant cells in mouse-model MPN. Transformed HSCs produce IL-β1, which damages sympathetic neurons and kill Schwann cells, leading to reduced CXCL-12 production and promoting mutant HSC proliferation [50]. Hypoxia and angiogenesis are two important traits of hematological malignancies, neovascularization induced by malignant cells provides increased nutrients and oxygen to supply the higher demand, but it also encourages the arrival of soluble factors that promote the survival, proliferation and chemoresistance of malignant cells. For example, vascular endothelial growth factor A (VEGF-A) secretion by transformed HSCs increases angiogenesis, but also stimulates its proliferation [51]. Under physiological conditions HSCs preferentially use glycolysis to avoid excessive ROS production and maintain quiescence. Malignant AML cells rely on different regulatory mechanisms to gain metabolic plasticity. Unlike their normal counterparts, malignant cells tend to have a high glucose uptake and glycolytic rate in the presence of oxygen, known as the Warburg effect (aerobic glycolysis) [52]. Although this results in low ATP yields, the Warburg effect is an essential anabolic mechanism that allows cancer cells to manage cell growth and division [53]. However, the field of tumor metabolism is extremely heterogeneous and specific fractions of malignant cells exhibit increased reliance on oxidative phosphorylation [54]. Some studies report that malignant cells uptake functional mitochondria from niche-forming cells using endocytic pathways to satisfy the greater demand for energy, which provides a survival advantage as it promotes ROS detoxification, resistance to chemotherapy and the aggressiveness of the disease [53].

Once transformed, BMN favors malignancy though different mechanisms. In some lymphoid malignancies such as CLL, B cell acute lymphoid leukemia (B-ALL) or mantle-cell lymphoma activation of survival and pro-inflammatory pathways, like the nuclear factor κB (NF- κB) pathway by altered BMN cells is necessary for malignant cell survival. This crosstalk is dependent on cell-cell contact and is also mediated by IL-1α and IL-15. Previously malignant cells modify BMN cells such as MSCs, inducing protein kinase C (PKCβ) expression [55]. Protection from excessive ROS is a different mechanism by which altered BMN protects malignant cells and promotes survival and chemoresistance. Altered BMN cells provide CLL or B-ALL cells with cysteine, used by malignant cells for production of glutathione for ROS detoxification, a key ability for survival, as chemotherapy effectiveness relies on ROS-induced DNA damage [56]. Metabolic reprogramming of malignant cells by altered BMN cells has also been reported. Altered BMN cells such as MSCs also promote immunosuppression and hamper the activity of effector lymphocytes by TGF-β, IL-10, prostaglandin E2 (PGE2) or arginase 1 or 2, and help to avoid attacks by malignant cells [57]. All these mechanisms combine to develop therapeutic resistance by malignant cells. The simple grafting of malignant cells onto specific sites could inhibit chemotherapy. CD44 has played a particularly important role in malignant cell interaction with the BMN in myeloid malignancies such as multiple myeloma, CML or AML [58,59,60]. CD44 binding with its E-selectin EC receptor mediates homing and grafting of malignant cells in CML [59] and enhances drug resistance in multiple myeloma [60]. Likewise, β1 integrins mediate adhesion to VCAM-1 or ICAM-1 molecules on stromal cells and induce CML cells adhesion to BMN. In AML, interaction between α4β1 integrin and VCAM-1 mediates chemoresistance towards activation of the NF-κB pathway in stromal cells [61]. Several studies report how the blockage of these interactions may help to sensitize malignant cells to conventional chemotherapy. In fact, the latest approaches to clinical treatment of hematological malignancies are combined therapies that not only attack malignant cells but also the altered BMN, or more specifically the crosstalk between supporting and malignant cells [62]. The importance of the environment in blood cancer pathogenesis is therefore undeniable (Figure 3). Effective in vitro modeling of hematological malignancies with BM homing, mainly myelomas and leukemias, goes through the inclusion into the model of different components from the native BMN.

## 5. Advances in BM Models

Engineering a BM analog would be important both for basic BMN research and for therapeutic strategies [63]. However, BM structural complexity, its spatially variable anatomy and the intricacies and dynamic character of its cellular interactions and regulation under physiological and especially under pathological conditions make BM and hematological malignancies particularly complicated to tackle for tissue and disease modeling. Despite the complications, TERM has developed tridimensional platforms attempting to overcome the limitations of conventional models, such as animal models or in vitro 2D culture. These platforms have been based on different approaches to provide structural support for HSCs and BMN-forming populations. BM is a complex machine with many different pieces (cellular components, ECM, soluble signals, oxygen level, tissue stiffness, etc.) working in an orchestrated manner to carry out its physiological roles. Different authors have chosen different approaches, in which some pieces conduct the niche’s in vitro mimicry that governs the spatial and temporal regulatory signals. Approaches include naïve cultures of hematopoietic progenitors in association with a biomimetic scaffolding material or in a co-culture with specific niche-forming cells through the more sophisticated BM-on-a-chip devices. Nonetheless, the development of a functionally effective model capable of showing BMN diversity and dynamism with the translational potential for disease modeling is still challenging. No single approach has been adopted as the standard in the field [63,64,65], however, different models have succeeded in reproducing some particular and restricted aspects of specific BM contexts. For example, the first attempts at reproducing cell-cell interactions started with the co-culture of HSCs and different BMN-forming cells, such as ECs, OBs or MSCs [66,67,68]. Cell-ECM interactions have been extensively incorporated into the equation by the introduction of natural tridimensional substrates or by polymeric synthetic substrates with different biofunctionalizations, all of them commonly presented in the form of scaffolds or hydrogels [69,70,71,72,73] (Table 2). Some of these approaches aim to improve the platforms for in vitro expansion of HSCs and ex vivo platforms for blood production [71,73]. Others aim to reproduce certain complex aspects of native BMN, such as compartmentalization and differential regulation of HSC fate [69,70,72]. All of them represent simplified bio-inspired set-ups that have undoubtedly contributed to increasing our basic knowledge of the hematopoietic compartment. However, their complexity is still far from assimilating BM complexity. Micro-scale systems or microfluidic perfusion chips have been used to mimic BMN complexity. Usually termed as BM-on-a-chip platforms, they include combinatorial approaches of fluid flow, biomimetic scaffolds and fine control of biochemical parameters (Table 3). These results are a step forward, as they succeed in mimicking functional compartmentalization, in providing knowledge that can be applied clinically or even reproduce the particular native behaviors of some blood cancers [74,75,76,77,78]. Nevertheless, what these different strategies do have in common is that they show that progress in the field relies on interdisciplinary approaches (Box 3). Although micro-scale models offer greater control, their design flexibility and significant result assessment are complicated. For the present authors, the reviewed models that most closely resemble BMN complexity rely on genetically modified humanized animal models [79], or the series of studies based on, bioengineered and humanized, mouse ectopically implanted microenvironments [78,80,81,82,83,84,85,86,87,88,89]. This raises the issue of whether animals should be included in the biological complexity, or if less complete models derived exclusively from human factors should only be used.

Box 3BM in vitro existing models.“Nowadays, the development of a functionally effective model with translational potential is still challenging (…). Different authors have chosen different approaches, and several models succeeded in recapitulating some behaviors of BM specific contexts (…) what these different strategies do have in common is that they show that progress in the field relies on interdisciplinary approaches (…)”.

Narrowing the context down to hematological malignancies, due to the intricacy of the role of BMN in tumor initiation and progression some authors use healthy BMN models with deliberately injected malignant cells to study pathological conditions, while others attempt to mimic pathological niches by including altered ECM or soluble tumorigenic environment [90,91]. For example, major efforts have been made to optimize complex BMN models for growing leukemia cells [92]. Highly porous scaffolds made from different biodegradable and non-biodegradable polymeric materials, such as poly (l-lactic-co-glycolic acid), polyurethane (PU), poly (methyl- methacrylate), poly (d, l-lactide), poly (caprolactone), and polystyrene coated with COL I or FN have been tested as models to study AML biology and treatment [93]. Inclusion of cell-cell interactions has also been considered: co-culture of BMN-forming cells with AML cells in a decellularized Wharton’s Jelly matrix (DWJM) showed higher resistance to chemotherapy than conventional suspension cultures and best resembled in vivo drug-resistance [94]. Different ECM components of ECM are known to play a role in inducing drug-resistance in different tumors: COL gel cultures induce drug resistance in different tumors; lymphoma or myeloma cells adhering to FN acquired resistance to mitoxantrone or dexamethasone, respectively, and HA is associated with drug resistance in leukemia [95,96]. Different authors have selected these biomolecules as scaffolding for in vitro reproduction of drug resistance [92]. For example, COL, FN and HA are present in DWJM scaffolding in the above-mentioned study on leukemic cells, generating a simplified but rationally designed platform for the study of ECM-induced drug-resistance [92,94,97]. Multiple myeloma (MM) is a hematological neoplasia in which the BMN’s role in disease progression via elevated proliferation, migration and CAM-DR has been extensively reported [98]. Simplified TERM approaches contributed to this knowledge: co-cultivation of MM cells with BM-MSCs revealed that these cells co-modulated their phenotype and that BM-MSC secretomes and microvesicles (MVs) participate in this crosstalk [99,100,101]. MM cell lines cultured on decellularized ECM from normal donors (ND) or MM patients’ BM-MSC recently showed that MM-MSCs’ ECM promotes MM cells’ MAPKs/translation initiation-dependent proliferation and migration, while normal donors’ ND-MSCs’ ECM has the opposite effect [91], an interesting result that supports the use of transformed in vitro conditions for modeling hematological malignancies. The more sophisticated platforms developed as healthy BMN have been used to study the behavior of malignant MM cell lines and effectively succeeded in reflecting resistance to conventional drugs, the main disadvantage in current MM clinical management [76,102].

## 6. Unresolved Questions in Modeling BM and Non-Solid Tumors

As the review delves into the advances in modeling healthy or malignant BM, some basic questions regarding the main design principles of TERM approaches should be considered. The physiological BMN and the transformations leading to or sustaining blood tumors are extremely complex and it will be a major challenge for TERM to faithfully reproduce these artificially in a BM model. In our view, cells must raise the artificial BMN from a provided substrate, which should be as biomimetic as possible, and from the presence of the appropriate environmental conditions (oxygen, fluid flow, physico-chemical stimuli…) (Box 4). The promising approach proposed by Torisawa et al., consisting of an artificial substrate engineered in vivo in a biomimetic BM model for in vitro HSC culture, strengthens the idea that the cells themselves are the last-resort architects of model complexity. However, this model is implanted in mice and does not closely mimic the human situation. For more humanized models, TERM must propose alternative methods of cell-guided niche formation. Some of the critical issues of these approaches to modeling blood cancers are examined in this section (Figure 5).

Box 4Unresolved questions in modeling BM and non-solid tumors.“The physiological BMN and the transformations leading to or sustaining blood tumors are extremely complex and it will be a major challenge for TERM to faithfully reproduce these artificially in a BM model. In our view, cells must raise the artificial BMN from a provided substrate, which should be as biomimetic as possible, and from the presence of the appropriate environmental conditions (oxygen, fluid flow, physico-chemical stimuli…).”

### 6.1. ECM Dynamics and Remodeling

ECM is a highly dynamic structure continuously undergoing controlled remodeling [103]. In vivo, cells rebuild and remodel ECM constantly through synthesis, degradation, reassembly, cross-link or chemical modification of their different components [104] and resultant biophysical, mechanical and chemical ECM characteristics influence tissue homoeostasis [105]. ECM remodeling is therefore an important mechanism helping to regulate cell differentiation or the establishment and maintenance of stem cell niches, angiogenesis, bone remodeling or wound repair [104]. Deregulated ECM remodeling is associated with different pathological conditions: abnormal ECM deposition and increased stiffness have been broadly linked with solid tumors [106,107]. In tissue engineering approaches, when adherent cells are seeded on 3D substrates (or on 2D surfaces) preexisting adhesion ligands from substrate functionalizations or serum protein deposition are essential for initial cell attachment, after which the cells remodel and secrete their own ECM. This inherent ability of ECM remodeling to control cell behavior has also been reported and exploited in in vitro 3D models [108,109]. In these approaches, both stiffness and degradability of the polymeric substrates are critical design variables, as metalloproteinases or other enzymes are commonly used by cells for ECM degradation, and natural or engineered proteolytically degradable materials have been extensively used in biomimetic approaches. Other groups have also proposed material-driven ECM remodeling. Salmerón-Sanchez et al. found that simple FN adsorption onto poly(ethyl acrylate) surfaces triggered FN organization in a fibrillar network similar to cell-assembled matrices [110]. These studies provide evidence of the cells’ ability to direct ECM remodeling in vitro and to enhance this process towards biomimetic ECM configurations. However, it has also been reported that cells embedded in artificial matrices require externally-imposed parameters, including matrix stiffness, which affect cell-mediated ECM remodeling by altering ECM regulatory genes [111]. This means that the initial matrix provided by the model will always guide cell behavior and influence cell-mediated ECM remodeling, perhaps reshaping the native output of this process. Therefore, the key question in hematological malignancies modeling is whether patient-derived malignant cells are able to can carry out this process or induce other cells in the model to do so.

### 6.2. Vascularization of the Model

To answer the question of how far cells can reconstruct native BMN complexity from an artificial substrate we should consider not only cell-mediated ECM remodeling but also the cells’ ability to create complex structures, such as vasculature, or reproduce the anatomical traits essential for tissue function, like compartmentalization. Vascularization of the model seems essential, as native BM blood vessels actively contribute to functional differentiation into subniches by their different BMN-forming cell populations and generating different biochemical gradients. In vitro vascularization is by itself a wide research field. Vascular networks have been generated in vitro by different approaches to study angiogenesis, vasculogenesis and cancer metastasis [112]. Vasculogenesis is the process by which early capillary-like networks form in vivo during adulthood through recruitment of endothelial progenitor cells (EPCs) from BM [113]. Following the formation of a primary network, expansion and sprouting occur from existing vessels by the process termed as angiogenesis, which is key in solid tumor growth, and activated by hypoxic environments or shear flow [112]. This mechanical stimulus also modulates the process of arteriogenesis, a maturation step that contributes to arterio-venous differentiation [113]. Many in vitro models are now enabled by microfluidic-based techniques and flexible polymers such as PDMS that can produce well-defined micro-scale geometries. One of the important aspects dealt with in this review is that in vivo vascular organization depends on the surrounding microenvironment, as seen in vessels aligned parallel to muscle, radially in the retina and highly branched in the lungs [112]. The in vitro patterning of the substrate or the composition of ECM mimicry used to develop the micro vessels influence their organization [114], even systemic factors, such as media flow, may affect network morphology and function [115]. Two options arise as dominant strategies for vascularization of engineered tissues in vitro: scaffold-based strategies using naturally derived and synthetically generated tube-like structures for guided vessel formation vs. the naturally formed cell-based strategies that rely on endothelial angiogenesis and vasculogenesis to form perfuseable networks [112]. ECs, EPCs, human dermal microvascular endothelial cells (HDMEC) or human umbilical vein endothelial cells (HUVECs), among others, have been used in in vitro angiogenesis and vasculogenesis models, always with a supporting ECM component and proper soluble environment to enhance their attachment and function [112,113,116]. However, as MSCs and other stromal cells have been shown to interact with endothelial lineage cells during neovascularization, MSCs have been co-implanted with human ECs, improving vascular tissue formation [117,118]. The most successful models of healthy or malignant BM reviewed in this article in terms of vascularization are those in which complexity has been engineered by implantation in vivo in mice. Although the human vasculature structure and perivascular BMN can be resembled in humanized structures implanted in mice [80], these approaches are again limited by their uncertainty as to the specific role of mouse vasculature and the cytokines supplied by the mouse system to these humanized implanted structures. Focusing on non-solid tumors, different authors have reported altered angiogenesis in association with different hematological malignancies. Laroche et al. reported that the number of arterioles and capillaries increased in myeloma from its initial stages according to the gravity of the disease [119]. Increased BM angiogenesis and increased levels of associated factors, such as VEGF, have also been described in relation to AML, ALL, B-CLL or CML, among others, leading to the development of therapeutic applications based on disruption of this mechanism [120].

The remaining question concerning vascularization is thus whether there is an effective animal-independent cell-guided way to biomimetically vascularize human engineered models, or in its absence, whether it is preferable to sacrifice the complex vascular biomimetic networks and replace them with gradients and an unstructured or artificially conditioned cellular and soluble factors from vessels [68] in order to maintain fully human models. This is the case of in vitro models of hematological malignancies such as the MM model in De la Puente et al., which fully preserve humanization and recapitulate the polarized BM niche structure with the generation of biochemical gradients, showing a more hypoxic vs. a more normoxic niche with more endothelial cells (Figure 4c) [76,121]. The present authors believe that recent advances in the field of in vitro vascularization, particularly the intention to provide better support to organ-on-a-chip approaches, will sooner or later contribute to the compatibility of human engineered vessels with native-like characteristics.

### 6.3. Compartmentalization

The more naïve bio-inspired setups summarized in Table 2 and some of the approaches in Table 3 contain several models that represent compartmentalization as a key factor in inducing mimicry. Diverse methods are used to create different functional zones, e.g., including structured compartments [21], differential matrix regions such as suspension vs. solid compartment [69,73], co-culture of different cells [70] or generation of biochemically different areas by gradients [72]. However, the combinatorial approaches are the most commonly used [74,75,76]. When moving forward towards the reviewed cell engineered approaches [65,77,78] it appears that in vivo-mediated vascularization, native ECM remodeling or including different cell types in the engineered models will indirectly lead to better functional compartmentalization than the initial design. Therefore, compartmentalization seems necessary to mimic functional BMN differentiation either by directly including it in the model or by indirect cell-mediated remodeling (Box 5). This has led the authors to consider the possible future role of bottom-up tissue engineering strategies in functional artificial BMN modeling, as this field aims to engineer complex tissues by the modular assembly of different living building blocks into customized architectures [122].

Box 5Compartmentalization.“Either by direct inclusion in the model by means of different design parameters or by indirect cell-mediated remodeling, compartmentalization seems necessary for functional differentiation of BMN”.

### 6.4. Stemness Maintenance Vs. Differentiation Balance

Another distinctive aspect of BMN is the balance between differentiation and maintenance of pluripotency, which is the key regulator of healthy and malignant BM niches (Section 5). Tumor cells interact with many cell types in their environment, especially in hematological malignancies, and in a trustworthy model these cell types have to coexist, as in in vivo conditions. How can pluripotent or multipotent cells be maintained at the same time as their differentiated counterparts and the production of trophic and regulatory factors? (Box 6). Ex vivo expansion of HSCs has been widely studied, mainly for its implications in hematopoietic cell transplantation (HCT) and generating red blood cells ex vivo. Uncommitted primitive cells are needed to efficiently repopulate the BM in HCT, which is still challenging since HSCs outside their niche tend to differentiate or become senescent [123]. The in vivo regulation of HSC self-renewal and differentiation is the reference pursued. Biomimetic co-cultures provide biological cues that liquid cultures, in the absence of stromal cells, barely replicate. Some studies maintain that the main factors that support HSC expansion rely on direct cell-cell interactions with different BMN-forming cells, while for others direct contact is not required, so that soluble factors play a key role in HSC fate. However, in agreement with [123], we believe that the interplay between soluble factors and direct ECM and BMN-forming cell interactions allows niche cues to efficiently regulate HSC fate. The choice of culture medium is crucial to determine HSC fate in vitro. Most approaches use supraphysiological concentrations of different HSC-supportive cytokines such as SCF, TPO, ANG-1 or IL-6 directly added to the medium and/or secreted by the BMN-forming cells in the model. ECM components, such as FN, COL, laminin or proteoglycans help to regulate HSC fate by binding the growth factors produced by BMN-forming cells, favoring cell co-localization and biological cues. Exploiting ECM’s ability to retain bioactive factors, Mahadik et al. bound SCF to a gelatin-based hydrogel to increase its bioactivity [124]. Diffusion control of nutrients, oxygen and cytokines by the ECM can also lead to gradients that could provide regulatory cues to HSCs [72]. Nevertheless, the presence of stromal cells seems to be key to promoting HSC self-renewal in vitro. Gottschling et al. showed that the presence of MSC was enough to ensure self-renewal, while the activation of β1-integrins by FN was not [125]. Different types of bioreactors have been used for HSC expansion ex vivo, including stirred tank suspension, perfusion chambers, fixed beds, airlift or hollow fiber reactors [123]. However, most of these rely on suspension cultures unable to maximize cell-cell and cell-ECM contact. The design of bioreactors coupled with biomaterial approaches has overcome this issue, such as that proposed in Sieber et al., whose 3-D co-culture model based on a hydroxyapatite-coated zirconium oxide scaffold with human BM MSC inserted in a microfluidic device was able to support long-term HSCs [74].

The issue of the differentiation vs. stemness balance is of interest not only for HSCs, blood and immune cells, but also for MSCs and osteolineage cells of varying maturity. Several approaches incorporate MSCs and pre-differentiated OBs in the model to cover this heterogeneity, although a dynamic balance between MSC expansion and OB differentiation would be of great interest. Maintaining MSC stemness ex vivo is also still a challenge, some microcarrier bioreactors have been used as large-scale production systems [126], although TERM approaches do not yet seem to have incorporated the dynamic balance between MSCs and OBs differentiation in BMN modeling. The initial steps have been taken, as there are many osteogenic scaffolds that efficiently generate well-differentiated OBs from initially seeded MSCs. However, the long-term co-existence of OBs, osteocytes and primitive MSCs seems to be difficult to regulate, as mature osteolineage cells seem to promote osteogenic MSC differentiation and lead to progenitor exhaustion [127]. Osteogenic media are commonly used in in vitro approaches, however, when considering co-cultures with HSCs or even with blood cancer cells such as MM cells, the use of osteogenic media should be rethought, since some of the soluble factors necessary for osteogenic differentiation of MSC could alter other cells included in the model. For example, dexamethasone is a commonly used anti-MM treatment (although the concentrations used in osteogenic media are 10 times lower [128]). The main problem concerning the use of specific conditioned media to promote specific phenotypic cell commitment in an ideal BM model is how to control the different effects of this media in different zones, thus allowing the coexistence of undifferentiated and mature cells. As this localized activation of cell differentiation does not seem to be possible by means of inductive media, we believe that TERM approaches have been shown to be efficient enough to induce differentiation in specific areas of the model and promote pluripotency in others. Anderson et al. reviewed specific aspects of scaffolding material design to rationally target MSC cell fate [129]. Mechanotransduction is the process by which MSCs turn an adherent stimulus into a cellular response able to determine cell fate. For cells to adhere to a synthetic surface, the material has to replicate an ECM motif or absorb ECM proteins, so that by controlling the ability of a material to allow cell adhesion, the subsequent activation of different signaling pathways that control stem cell fate can also be defined [129]. There are important design parameters for this purpose: material chemistry, stiffness and nanotopography are known as the material/surface interface “triangle”, as their variations control the scaffold’s interactions with MSCs by conditioning the formation of focal adhesions. Binding the cell through focal adhesions to adhesive ligands on the materials creates tension and activates signaling that controls cell behavior and stem cell fate. Chemical functionality can be used to produce high or low adhesive areas for the cells to respond to stiffness, which affects their ability to create tension through focal adhesions. Topography can present the adhesion ligands to the cells in either a favorable or unfavorable way, again affecting adhesion and subsequent tension and signaling [129]. The chemistry of the surface and its conditioning of matrix protein deposition, which in turn regulates the presentation of cell adhesion motifs to cells, can be finely tuned with techniques such as dip pen nanolithography (DPN) to apply a surface chemistry to a precise substrate on the nanometer scale. For example, Curran et al. set out to optimize an arrangement of dots of “chemistry” to manipulate MSC behavior by creating specific patterns of -CH_3_, -NH_2_, -CO, and -CO_2_. They found that the functionalized -CH_3_ surface maintained stem cell markers while -NH_2_ dots increased adhesion and osteogenesis [130]. Precise control of chemical functionality in 3D is still challenging, however ongoing progress in nanofabrication techniques promises to make significant contributions in the near future [129]. ECM topography in vivo presents a native composition that provides the cells with behavioral cues. in vitro, the topographical cues influence on stem cell behavior has also been proved and explored by many scientists as a tool to manipulate MSC fate. Dalby et al. developed MSC growth substrates with random or highly ordered patterns and observed that osteogenesis increased in the disordered patterns as efficiently as in inductive media [131]. However, these studies imply a 2D character and a “static” nature, in the sense that they include a configuration of the material oriented to promoting a single effect on the cell fate (self-renewal or differentiation). While the stem cell niche and the BMN are dynamic microenvironments in which the balance between stemness and differentiation is regulated by the demand. Next-generation materials able to support self-renewal and differentiation with spatiotemporal control have attracted significant interest in recent years. For example, stimuli-respondent materials in which a cytocompatible stimuli such as light triggers material changes leading to alterations in cellular behavior, together with advances and tridimensional implementation of microfabrication techniques, will address the future need for niche-mimicking materials [129].

Box 6Stemness maintenance vs. differentiation balance.“In hematological malignancies, tumor cells interact with many cell types (…). How to maintain in culture pluripotent cells at the same time as their differentiated counterparts? TERM approaches have shown to be efficient enough to, in a localized and differential manner, induce differentiation in specific areas of the model and promote pluripotency in others”.

### 6.5. Cell Culture Media Renewal and Composition

Although culture media renewal is usually considered as routine in in vitro cultures, apart from its role as a supplier of nutrients and externally selected regulatory factors, a common problem in both static and bioreactor cultures is that this renewal may remove secreted factors essential for the relationship of the different cells in the culture system. We hypothesize that the effect of reducing active component concentration by renewing media can interfere with physiological behaviour. The dynamics and time lapses required for these processes have not yet been clearly established. However, the effect of soluble factor dilution has been reported in particular applications; for example, spontaneous in vitro HSC differentiation was avoided by diluting secreted differentiation signals via proportional volume to cell number ratio [132]. Some studies on the use of conditioned media for several applications and culture duration for conditioned media generation from different stem cells found that culture conditions could vary from 16 h to 5 d [133]. The timelines of some mammalian cellular processes could help in understanding the timelines in which cells grow and communicate. For example in a human HeLa cell, diffusion of a protein across a cell will take 10 s in a 10 µm cell, while transcription of a 10 kbp gene takes approximately 10 min, and 1 min to translate a 300 aa protein [134]. This means that a cells’ ability to reconstitute the removed soluble factors from the culture media is probably within these orders of magnitude. Some cellular responses to different stimuli may also be in the same order and are thus probably altered to some extent by medium renewal, e.g., the on-switch of apoptosis in HeLa cells, which has been reported to take between 9 and 29 min [135].

### 6.6. Towards Personalized Medicine

Research has shifted from gold standard treatments for a given cancer to finding solutions for patient-specific cancer subtypes or, in other words, personalized medicine [136]. Drug effectiveness differs greatly between individuals, a cure for one patient can be ineffective or harmful to another. When modeling hematological malignancies, the challenge lies in reproducing cell-cell and cell-ECM interactions in a 3D environment to regulate the signaling pathways leading to drug-resistance. However, inter-patient heterogeneity has driven the need for personalized medicine as individual tumors have different gene expression profiles, tumor microenvironments and behavior even within the same cancer subtype [136]. The challenge of recreating niche interactions leading to drug resistance has advanced to reproducing inter-patient differences in the model (Box 7). Personalized cancer therapy has historically focused on profiling tumor DNA, RNA, or protein as molecular biomarkers to predict patient response. However, these methods have not been able to predict therapeutic response [136], while functional assays based on integrating tumor cells into chemosensitivity and resistance assays have become a complementary method [137]. In 2017, Snijder et al. evaluated the effect of ex vivo drug sensitivity screening on the treatment of patients with refractory hematological malignancies using the Pharmacoscopy automated immunofluorescence microscopy-based platform on 48 patients, 17 of whom received treatment guided by this approach. Comparison of the benefit of Pharmacoscopy-guided treatment with the effect of previous treatments in the same patients showed a marked improvement in progression-free survival with the former, providing evidence of the promise of drug-response profiling in haemato-oncology [138]. Similar studies have served as the proof-of-concept of how phenotypic screening approaches to different blood cancers such as AML [139] or MM [140,141] could improve the selection of the right drug for the right patient at the right time. Although these approaches are still simplistic in terms of BMN mimicry, some of them include co-culture with BM stromal cells and ECM components like COL [141]. However, the most important factor is that they have been the pioneers in introducing multidisciplinary 3D biomimetic models that reproduce tissue architecture to revolutionize the clinical management of cancer patients. Certain types of blood cancer such as MM would greatly benefit from these advances, as it is a treatable but incurable malignancy in which all the patients eventually relapse and the choice of their treatment now relies on clinical acumen instead of empirical personalized data [141].

Personalized drug resistance assays have thus shown promise, although they still remain extremely simplistic compared with BMN harboring cancer in vivo [136,142]. TERM engineered biomimetic models are rapidly progressing. Microscale models can provide unique functionality and controllability (e.g., enhanced spatial and temporal controls) and are emerging as practical tools to investigate tumor-stroma interactions [143]. Improvement of 3D substrate production techniques and their use for stem cell fate regulation [129,144], even on patient-derived ECM as scaffolding materials [14] and advancement in microscopy, flow cytometry and different evaluation techniques led the way to integrating patient samples into these in vitro models to assess therapeutic response in biomimetic devices and overcome the limitations of immortalized cell lines. The latter are highly selected populations that do not reflect the heterogeneous tumor genetic and functional variability. Patient-derived primary cells would overcome this disadvantage, although their use is hindered by issues of patient sample acquisition, variability and the difficulties associated with their culture. Integrating patient cells requires that part of the sample, which would ordinarily go to pathology, which is not always possible, is specially challenging in the case of hematological cancers in which extra mL of BM aspirate should be extracted. Secondly, it is difficult to transfer the sample from the operating theatre to the in vitro platform while maintaining cellular viability. The most questionable aspect of this process is that the sample usually needs processing, which introduces additional variations in tissue architecture, microenvironment and cell selection. In addition, cells often undergo changes when removed from their natural environment (primary cells easily become senescent, and have a limited availability and lifespan). These are well known unsolved problems that hamper the development, optimization and validation of new assay platforms [136]. We also consider that other issues not widely included in the literature produce uncertainty and complicate the incorporation of primary cells into hematological malignancy models, as there are questions regarding the timing of cell expansion before seeding into the models, types of cells and number or proportions of cells. For example, how long does it take for a tissue sample from a patient to have the required number of cells of all the relevant types in a culture to be able to carry out drug resistance tests and get statistically significant results? Does a BMN aspirate from a patient efficiently represent the proportions of key niche cells such as MSCs or HSCs?

Despite the progress made, it is undeniable that choosing the right cancer treatment is difficult because of limited tools, money and time [136]. Important steps have been taken, as some authors have convincingly resembled acquired drug resistance in vitro with cell lines [76,102], but some issues need to be addressed until optimized models allow routine patient-specific drug testing studies. Personalization means that the model must consider inter-patient differences in drug response. In the authors’ opinion interactions between tumor cells and the cellular and non-cellular components of the biological niche play a key role in determining these differences. This means that personalization is a key step towards obtaining the perfect blood cancer model. They also believe that the cells themselves, acting as the last-resort architects of the particular extracellular environment provided by the 3D biomimetic model, improve the inter-individual specificity of the biological niche and this cannot be artificially reproduced.

Box 7Towards personalized medicine.“The challenge lies in reproducing in a 3D environment the cell-cell and cell-ECM interactions which regulate the signaling pathways leading to drug-resistance (…). Personalization means that the model must consider inter-patient differences in drug response, thus the challenge lies in reproducing differences from patient to patient (…), in integrating patient samples in models to assess therapeutic response in biomimetic devices, and overcome the limitations of immortalized cell lines”.

## 7. Conclusions

This review combines information from different fields with the aim of providing an interdisciplinary view of the biological context and design principles for in vitro models of blood cancers with BM homing. The BMN is a complex cellular and non-cellular microenvironment and an important factor in tumor progression and drug resistance. The use of more physiologically relevant cultures would improve in vitro model prediction of drug response and should therefore be further explored. This model would ideally incorporate all the tumor components and microenvironment, with a trade-off between complexity and physiological relevance with reproducibility, ease of use and cost. Some authors recommend the simple incorporation of specific elements of the in vivo environment into models to better evaluate the response of a given therapy. However, TERM strategies are now available and there are a wide range of aspects of the ideal BM model in which the degree of mimicry should be further explored, such as vascularization, architecture or cell fate regulation by the biomaterials. One of the challenges still to be addressed is the development of complex models based entirely on human cells, which would provide a powerful platform for basic research and clinical translation. In our view, most attempts at modeling blood cancers have involved “deconstructing rather than reconstructing” the complexity of native BM. Although this has limited the contributions, at the same time it has made it possible for the models to advance clinical and basic research, and more dynamic and biomimetic TERM strategies are now emerging for blood cancer modeling.

## Figures and Tables

**Figure 1 ijms-21-05747-f001:**
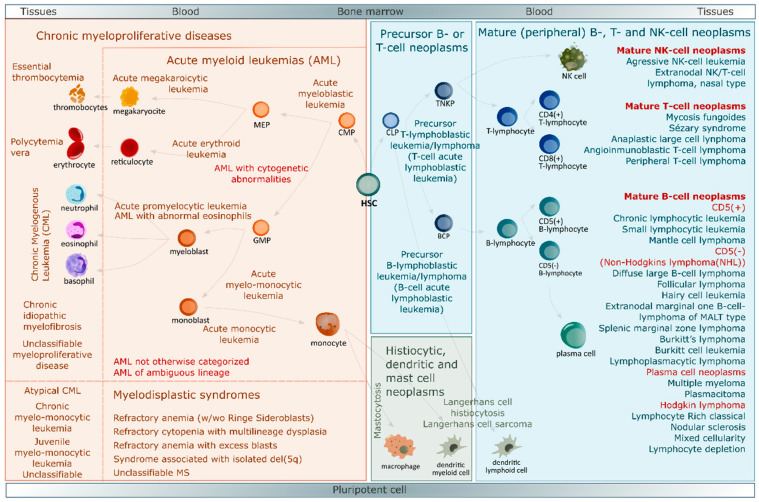
Representation of BM hematopoiesis together with the simplified WHO classification of neoplastic diseases of the hematopoietic and lymphoid tissues [28,29]. Blue box contains lymphoid neoplasms, orange box myeloid neoplasms and green box rare histiocytic, dendritic or mast cell neoplasms. Abbreviations: del, deletion; HSC, hematopoietic stem cell; CMP, common myeloid progenitor; CLP, common lymphoid progenitor; MEP, megakaryocyte-erythrocyte progenitor; GMP, granulocyte-monocyte progenitor; TNKP, T-natural killer cell progenitor; BCP, B cell progenitor; NK, natural killer.

**Figure 2 ijms-21-05747-f002:**
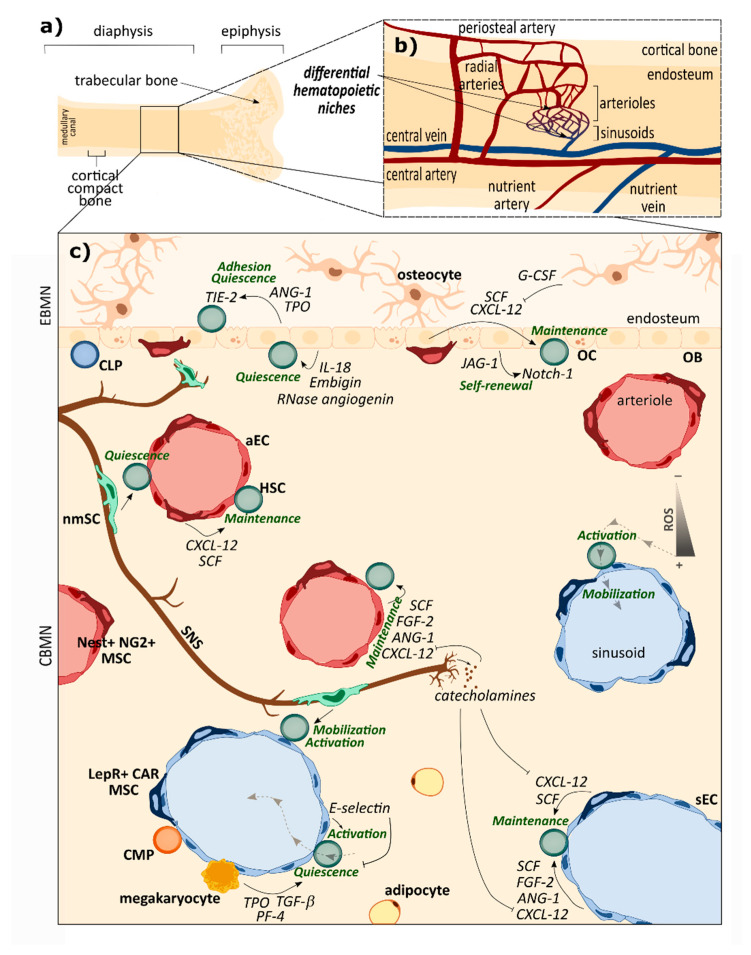
Scheme of bone vasculature structure and healthy BMN. (**a**,**b**) BMN location and vasculature architecture. Arterial vessels that penetrate cortical bone merge and form the central artery. Arterioles branch from the central artery toward cortical bone and anastomose with the sinusoid, which then connects with the central vein in bone surface proximities. The vascular structure of the medullary canal in bones contributes to spatial delimitation of the BM HSC sub-niches. (**c**) Representation of cell populations (bold text) and their locations in BMN; main cell-cell interactions by direct contact and soluble factors and their role in HSC homeostasis (green) are detailed. Abbreviations: EBMN, endosteal bone marrow niche; CBMN central bone marrow niche; ROS, reactive oxygen species; HSC, hematopoietic stem cell; CMP, common myeloid progenitor; CLP, common lymphoid progenitor; aEC, arteriolar endothelial cell; sEC, sinusoidal endothelial cell; LepR^+^ CAR MSCs, leptin receptor expressing cells and abundant reticular mesenchymal stem cells; Nest^+^ NG2^+^ MSCs, nestin and neural glial antigen expressing mesenchymal stem cells; SNS, sympathetic nerve fiber; nmSCs, non-myelinating Schwann cells; OBs, osteoblasts; OCs, osteoclasts; CXCL-12, CXC motif chemokine ligand 12; SCF, stem cell factor; ANG-1, angiopoietin 1; TIE-2, angiopoietin receptor; IL-18, interleukin 18; FGF-2, fibroblast growth factor 2; TGF-β, transforming growth factor beta; TPO, thrombopoietin; PF-4, platelet factor 4; JAG-1, Jagged-1; G-CSF, granulocyte-colony stimulating factor.

**Figure 3 ijms-21-05747-f003:**
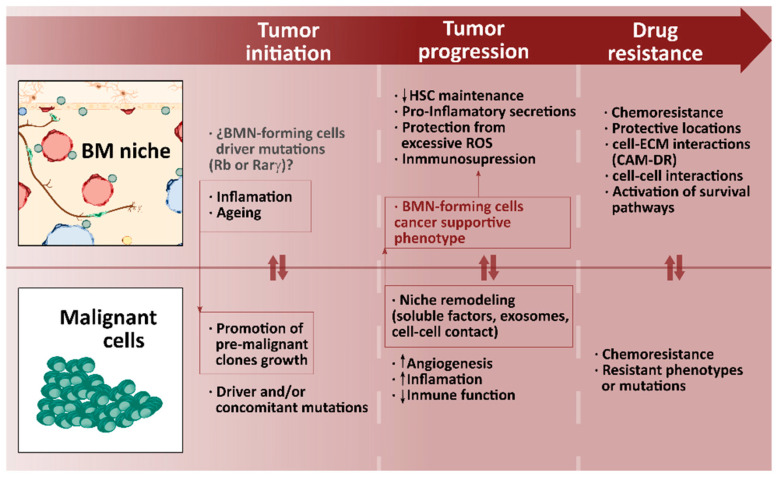
Diagram of malignant cells and BM niche interplay guiding tumor initiation, progression and acquisition of resistance to therapies. Abbreviations: BM, bone marrow; BMN, bone marrow niche; Rb, Retinoblastoma; Rar_γ_, Retinoic acid receptor γ; HSC, hematopoietic stem cell; ROS, reactive oxygen species; CAM-DR, cell adhesion-mediated drug resistance.

**Figure 4 ijms-21-05747-f004:**
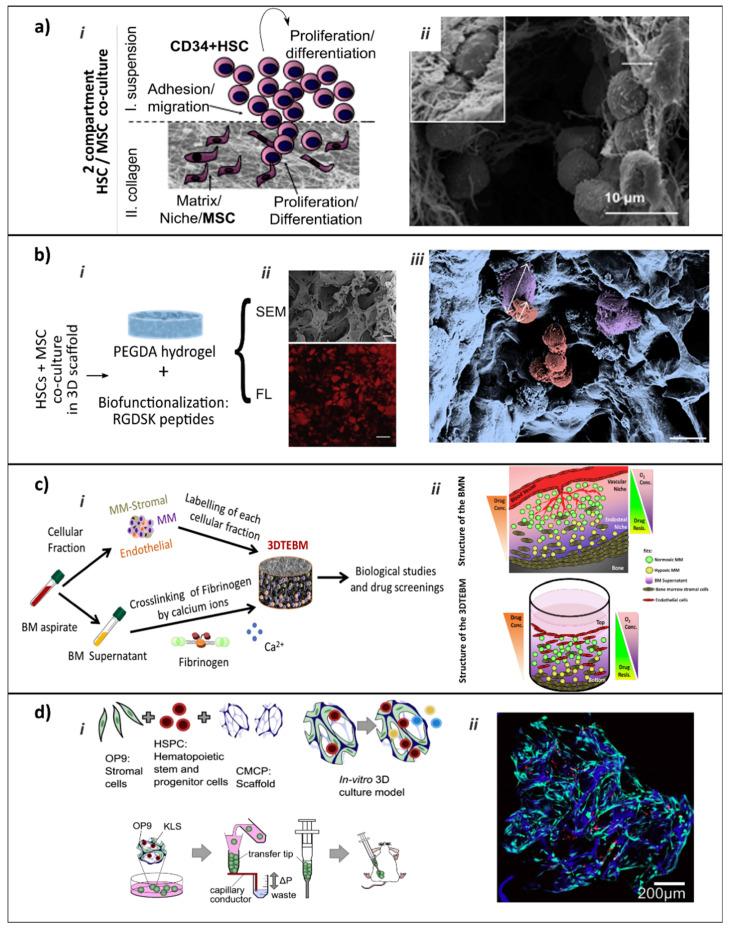
TERM models of healthy BMN. (**a**) Co-culture of human HSPCs with MSCs in 3D COL hydrogel dissects two sub-populations of HSPCs. (i) Scheme of the 2-compartment system. (ii) HSC morphology by SEM at day 14 (adapted by kind permission of [69]). (**b**) Co-culture of HSPCs with mesenchymal stromal cells in PEGDA hydrogel coated with adhesive peptides mimicking trabecular bone. (i) Scheme of the approach. (ii) Cross-sections of hydrogels, SEM images (top panel) and fluorescence micrographs (lower panel) with Alexa Fluor 647-labeled BSA in red as tracer molecule to reveal interconnectivity of the pores. Scale bars, 20 µm (SEM) or 100 µm (FL). (iii) Pseudo colored SEM micrograph of a hydrogel seeded with primary MSC-BM and HSPCs. Purple: adherent MSC-BM, 28 µm in size; red: HSPCs, 12 µm in size. Scale bar, 20 µm (adapted by kind permission of [70]). (**c**) In vitro 3DTEBM used for culture of BM aspirates from MM patients. (i) Scheme of the approach; cultures were developed through cross-linking of fibrinogen with calcium; numerous cellular components, including MM cells, MM-derived stromal cells and endothelial cells were incorporated. (ii) Illustration of the BMN and the 3DTEBM with oxygen and drug concentration gradients (adapted by kind permission of [76]). (**d**) Cryogel-based COL coated carboxymethylcellulose scaffold seeded with murine BMN-forming cell line OP9 to generate a living, injectable stroma supportive for hematopoiesis, and with murine HPSCs, seeded scaffolds used as microcarriers for in vivo culture in mice (i). (ii) Fluorescence image of CCM seeded scaffold, blue: Hoechst (scaffold); green: GFP (OP9 stromal cells); red: HSPCs (adapted by kind permission of [65]). Abbreviations: SEM, scanning electron microscopy; HSPC, hematopoietic stem and progenitor cell; PEGDA, Poly (ethylene glycol) diacrylate; HSC, hematopoietic stem cell; MSC, mesenchymal stem cell; BM, bone marrow; 3DTEBM, 3D tissue engineered bone marrow; MM, multiple myeloma; COL, collagen.

**Figure 5 ijms-21-05747-f005:**
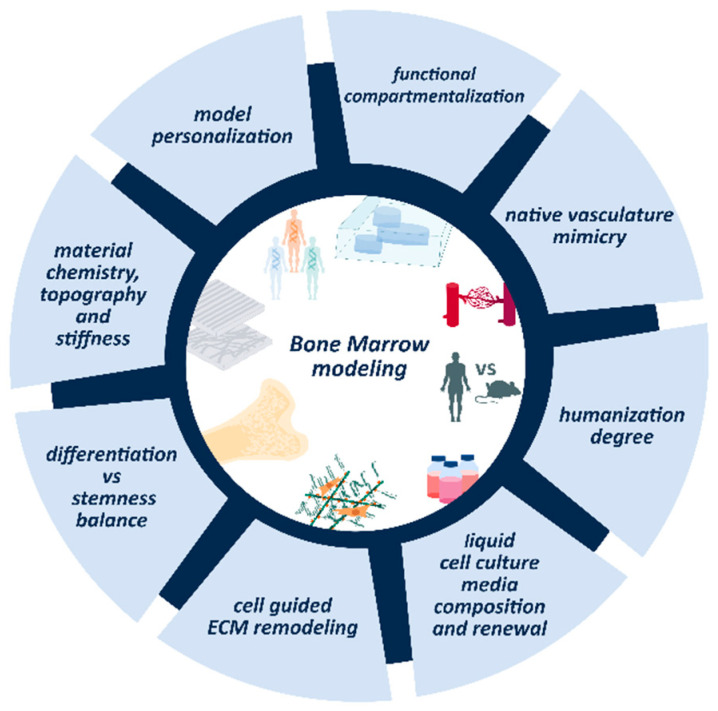
Scheme of critical issues that should be addressed by TERM blood cancer model approaches.

**Table 1 ijms-21-05747-t001:** Summary of different TERM approaches for modeling solid tumors.

Reference	Type of Model	Material and ECM Mimicry	Tumor Type (Cell Line/Cell Source)	Application or Description	Conclusions
[15] 2016	Scaffold-free. Spheroids.	Methylcellulose (for spheroid formation).	Colorectal cancer. (Cell line CaCo2).	Comparative study of tumor related pathway signaling in planar vs. spheroid culture.	Spheroids present diminished AKT–mTOR–S6K signaling. mTOR activity and crosstalk between AKT–mTOR–S6K signaling and the MAPK pathway is altered in 3D cultures. Spheroids present in vivo-like mTOR-S6 signaling gradients.
[19] 2017	Scaffold-free. Patient-derived organoids (tumoroids).	Basement membrane extract. Matrigel.	Primary liver cancer. (Cell source: patient derived cells from 3 tumor subtypes; hepatocellular carcinoma, cholangiocarcinoma and combined).	Validation of patient-derived organoids as preclinical personalized cancer model. Identification of potential prognostic biomarkers and patient-specific drug sensitivities.	Primary liver cancer-derived organoids preserve features of native tumor and subtype in vitro. Tumorigenic and metastatic potential are preserved in vivo (xenograft implantation). Amenable systems for biomarker identification and drug testing. Identification of the ERK inhibitor SCH772984 as a potential therapeutic agent.
[21] 2017	Scaffold-based.	Poly-ether- urethane foam.	Breast cancer. Bone metastasis model. (Cell source: human ADSC and MCFS).	In vitro model to recapitulate the metastatic spreading of breast cancer in bone tissue.	Importance of osteoblasts in mediating adhesion and growth of breast cancer cells. MCFS proliferate and form aggregates in the co-culture biomimetic model. MCFS affects Ca and P deposition in the bone mimetic tissue.
[22] 2019	Hydrogel-based. 3D bioprinting.	Gelatin methacryloyl UV cross-linked.	Bladder cancer. (Cell lines: 5637 and T24).	Development of a 3D environment for tumor formation and chemotherapy response characterization.	3D cultures showed higher cell proliferation and cell-cell interactions (E and N-cadherin expression). 3D cultures showed diminished response to rapamycin and Bacillus Calmette-Guérin.
[24] 2019	Scaffold-based. Spheroids. Microfluidics.	Polystyrene scaffold. Microfluidics. Poly-l-lysine and laminin-1 coating.	Breast and lung carcinoma. (Cell source: 3D tumor spheroids displaying CSC-like features from breast (MCF-7) and lung (A549) cancer cell lines).	Reproduction of the adhesion process of CSC to a target tissue by using a 3D dynamic cell culture system.	Development of a 3D dynamic model to study metastasis processes, such as formation of premetastatic niche and attachment of circulating tumor cells.
[23] 2019	Tumor-on-a-chip. Microfluidics.	PDMS. Matrigel.	Colorectal cancer. (Cell source: human colon cancer cell line HCT-116 and HCoMECs).	In vitro 3D microfluidic cell culture for studying onco-nanomedicine efficacy.	Validation of model with tumor core supported by adjacent microvasculature compatible with real-time image analysis, gradient-like response on cancer cells, supports a stable and viable co-culture of HCoMECs and HCT- 116 and gene expression analysis.

Abbreviations: ADSC, adipose derived stem cells; MCFS, breast cancer derived tumor initiating cells; CSC, cancer stem cells; PDMS, poly (dimethyl siloxane); HCoMECs, primary human colonic microvascular endothelial cells.

**Table 2 ijms-21-05747-t002:** Summary of different TERM approaches for modeling BMN interactions.

Reference	Factors of Mimicry	Cellular Component	Biomaterial	Achievements
[69] (Figure 4a)	Cell-cell and cell-ECM interactions. BMN Compartmentalization.	Human MSCs from UC or BM. HSPCs from UC blood.	COL I /III based hydrogels.	3D co-culture system resembles the EBMN and dissects two sub-populations of HSPCs: (I) highly proliferative with tendency to lineage commitment and (II) with clonal expansion and immature phenotype with self-renewal and repopulation capacity.
[70] (Figure 4b)	Cell-cell and cell-ECM interactions.	Human HSPCs from UC blood. BMN-forming cells: MSCs (from BM and UC) and OBs cell line CAL-72.	PEGDA hydrogel mimicking trabecular bone. Adhesive peptides.	Co-culture showed more pronounced positive effect of MSCs on preservation of HSPCs stemness in 3D than 2D. Bio-functionalization offers adhesive sites, supplemented medium provides soluble factors, MSCs reflect the supporting stromal cell compartment.
[71]	Cell-ECM interactions. BMN Compartmentalization.	Human CD34^+^ cells from adult peripheral blood.	PU scaffold with honeycomb structure.	Compartmentalized scaffolds allow harvest HSCs across longer periods. Continuous egress of cells with an erythroid progenitor phenotype over a 28 days period. Maintenance of CD34^+^ population, while facilitating egress of increasingly differentiated cells.
[72]	Cell-cell and cell-ECM interactions. Biotransport.	Murine BM derived Lin^−^Sca1^+^cKit^+^ (LSK) sub-fraction and Lin^+^ BMN-forming cells.	Cell-laden COL I hydrogels with varying densities.	Co-variation of hydrogel diffusivity and BMN-forming cell density controls HSCs proliferation vs. differentiation by varying autocrine vs. paracrine signaling. Biotransport limitations in 3D models as critical design element.
[73]	Cell-ECM interactions. Soluble factors improved presentation.	Murine ckit+ enriched HSCs cells from BM.	PVA. FN as 2D coating for HSCs retention. Soluble TPO and SCF.	Ex vivo platform for long-term HSCs expansion. Affords 1-month expansion of functional HSCs. Cultures derived robustly engrafted in recipients without requirement for toxic pre-conditioning, suggesting new approaches for HSC transplantation.

Abbreviations: ECM, extracellular matrix; BMN, bone marrow niche; BM, bone marrow; MSC, mesenchymal stem cell; UC, umbilical cord; EBMN, endosteal bone marrow niche; HSPCs, hematopoietic stem and progenitor cells; OB, osteoblast; PEGDA, poly (ethylene glycol) diacrylate; PU, polyurethane; HSC, hematopoietic stem cell; FN, fibronectin; TPO, thrombopoietin; SCF, stem cell factor; PVA, polyvinyl alcohol.

**Table 3 ijms-21-05747-t003:** Summary of different complex microscale systems or BM-on-a-chip approaches.

Reference	Approach	Achievements
[74]	Hydroxyapatite-coated ceramic cancellous bone mimicking scaffold, pre-culture with primary BM-MSCs inducing ECM deposition and factor secretion. Co-culture with HSPCs from UC blood developed in the MOC platform [69].	Long-term culture of HSPCs. MSCs generated microenvironment conducts HSC maintenance. HSPCs remain their native state after 4-weeks culture in dynamic conditions in the perfused MOC and retain multi-lineage differentiation potential. MOC platform allows co-culture with different organoids in adjacent chambers
[75]	Bone-like ceramic scaffold, functionalized by human stromal cells and by the ECM they deposited during perfusion culture in bioreactors.	Perfusion-based bioreactor system, partially recapitulating structural, compositional and organizational features of EBMN. Support of HSPCs maintenance and expansion in vitro with preserved multilineage reconstitution potential. Functional compartmentalization. Possibility to exploit the system for study BMN with customized molecular signatures.
[76] (Figure 4c)	Micro-scale 3DTEBM cultures derived from the BM supernatant of MM patients. Different BM cellular components (MM cells, BMN cells, and ECs). Cross-linked fibrinogen scaffold.	3DTEBM cultures allowed proliferation of MM cells, recapitulated their interaction with the microenvironment, recreated 3D aspects of BMN (such as oxygen gradients), and induced drug resistance in MM cells.
[77]	PDMS hollow compartment with a COL I gel containing bone-inducing DBP, BMP2 and BMP4, implanted subcutaneously in mouse. Posteriorly explanted and maintained in vitro in a microfluidic device for 4 or 7 days. Functionality and responsiveness of the tested by exposing it to γ-radiation to determine whether this method could be used as an in vitro model for radiation toxicity.	Formation a cylindrical disk of cortical and trabecular bone containing marrow with a hematopoietic cell composition nearly identical to that of natural BM. Presence of key cellular and molecular components of BMN. During posterior 1-week in vitro culture retained morphology and molecular patterns, enabled maintenance of a significantly higher proportion of long-term HSCs while effectively maintaining distribution of mature blood cells. Mimicked physiological response to clinically relevant doses of γ-radiation.
[78]	Generation of humanized heterotopically localized bone organoid, “ossicles”, recapitulating normal BMN morphology and function. Ossicles formed in-situ by BM-MSCs ectopically implantation in mice. HSCs can subsequently be transplanted into the ossicle. Transplantation of normal and malignant HSCs.	Robust and reproducible in vivo methodology to study human normal and malignant hematopoiesis in a physiologic setting. Effectively engraftment of primary patient-derived AML and myelofibrosis cells in mice. Although bone, cartilage, and MSCs within the ossicle are of human origin, the vasculature is mouse derived. Limited applicability of the model to human specific questions.
[65] (Figure 4d)	Cryogel-based COL coated carboxymethylcellulose micro-scaffold seeded with murine BMN-forming cell line OP9 to generate a living, injectable stroma supportive for hematopoiesis, and with murine HPSCs. Seeded scaffolds act as microcarriers, enabling culture *in vivo*, when implanted ectopically in mice for vascularization.	Scaffolds promote hematopoietic cell proliferation over time, amenable to live, high-resolution imaging. Co-culture on chemically defined scaffold microcarriers. Simple and scalable. No exogenous cytokine supplementation. Stromal and hematopoietic cells able to survive in vivo for 12 weeks, showing incorporation into the native tissue via de novo vascularization.

Abbreviations: BM, bone marrow; MSC, mesenchymal stem cell; ECM, extracellular matrix; HSPCs, hematopoietic stem and progenitor cells; UC, umbilical cord; MOC, multi-organ-chip; EBMN, endosteal bone marrow niche; BMN, bone marrow niche; 3DTEBM, 3D tissue engineered bone marrow; MM, multiple myeloma; EC, endothelial cell; COL, collagen; HSC, hematopoietic stem cell; DBP, demineralized bone powder; BMP2, bone morphogenetic protein 2; BMP4, bone morphogenetic protein 4; AML, acute myeloid leukemia.

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
