# Peer review of "In Vitro Modeling of Non-Solid Tumors: How Far Can Tissue Engineering Go?"

_ijms, 2020, doi:10.3390/ijms21165747_

Round 1
Reviewer 1 Report
The present review is a deep and complete bibliographic summary of what is known in the field about recreating hematological malignancies with in vitro strategies and how all components of the niche play an important role for modeling these diseases.
The manuscript is very well written, easy to understand and includes all necessary concepts to get into the topic. It also contains many relevant references and updated previous reviews in the field.
I have some comments for the authors to help them improve the manuscript:
-In line 299 it is mentioned that AML cells and other malignant cells depend on oxidative phosphorylation for survival. This is not completely true since most of tumor cells rely more in anaerobic glycolysis to obtain their energy, even when it is less efficient for obtaining energy than oxidative phosphorylation. This is known as the Warburg effect. Authors should mention this and clarify this issue. Also it has to be taken into account that within AML cells there are different populations and the metabolisms of the leukemic stem cells is not the same as the blasts cells.
-Since the great value of a review is to be able to find all references supporting what it is mentioned in the text, I missed many primary references describing the original results and data of what is described along the manuscript. In general the references mention other previous reviews but I think it will increase the relevance of this manuscript if some original references are also included. For example, review 33 is many times mentioned along the text and sometimes also the original papers should be referenced.
-Figure 4 includes a lot of information and very small text that is difficult to read. Maybe it would be more informative if authors just keep the scheme of the different strategies used in the 4 studies and include the reference for readers who want to know the specific results and details.
-I suggest to include in figure legends and tables the meaning of all abbreviations appearing in the figure, even when they are already mentioned in the text. This will make easier to understand for the reader.
-In line 159, the format of the reference should be consistent with the rest of the manuscript ([33] instead of (Mendez-Ferrer et al 2020)).
-In figure 2, it is difficult to understand what is written in green, red and blue. It would be convenient to make it clearer.
-In line 260, two studies are mentioned, please provide the reference.
-In line 264, shouldn’t be a period after reference 33.
-In line 416 there is a typo: os should be of.
-Line 462: never say never….
-Line 495: there is a typo: Salmeó n-Sánchez should be Salmerón-Sánchez
-Line 519: after ref 80 should be a period instead of a comma.
-Abbreviations: Since there are many abbreviations along the manuscript it would be much helpful for the reader and easier to find if there would be in alphabetical order.
-References: Please check all references because there are some with missing information (example: ref 1, ref 33, ref 35, ref 70, ref 100, ref 101).
Additionally, some references lack the doi number or they are in inconsistent format.
Reviewer 2 Report
The review by Clara-Trujillo et al gives a nice and comprehensive overview of current developments in the field of developing in vitro models to study normal and malignant hematopoiesis. I find the review well structured and very informative. The authors have provided a thorough overview of various in vitro bone marrow models and added their own comments and views on their development and future aspects. They also have included figures and tables that very nicely summarise the points they want to discuss.
However, I have a few comments on issues that I believe the authors should address in order to improve the quality of the manuscript:
1) The review is mostly based on citations of other reviews. Indeed, almost half (and possibly more) of the references are reviews or commentaries, instead of original research papers. I believe that citing a review is fine when making general statements or when giving an overview of a field that is not the central point of the manuscript (e.g. the short description of hematopoiesis on page 5 of this review). Nevertheless, when very specific findings are mentioned the original research that led to those findings should be cited, rather than another review which discusses these findings. This ensures that proper credit is given to the researchers who published these findings, and also helps the reader to easily find the original publication if they wish to do so. This is a very important function of a review.
The authors on several occasions cite reviews instead of the original papers. This is most striking in section 4.2, in which a lot of information is given but only 5 references are included (33, 36, 38, 39 and 40), all of which are reviews. I recommend that the authors revise the references so that they add the original research publications which first showed the specific findings that they mention. This applies to the whole manuscript, but at least to the specific findings mentioned in the following lines: 260-265; 267-268; 274-275; 281-284; 284-285; 286-289; 290-292; 295-297; 300-301; 307-308; 313-314; 320-322. The authors should add those original research publications.
2) I would recommend a thorough check of the language, if possible with the help of a native speaker. This refers not so much to mistakes or mis-spellings, but rather to improving the clarity of sentences, which is often an issue in the manuscript.
3) Minor point: it is not clear if the data in Figure 4 c (ii) are from a published study or they are the authors' own data. If it is the former, then the publication should be mentioned. If it is the latter, then the authors should add some more information about the experiment, such as the composition of the culture media used (any cytokines added?) and how many replicates are shown. Moreover, it would be nice to enlarge this very busy (and informative) figure, if possible.
4) Minor point: in Figure 2, "CD5(-) T-lymphocyte" should be "CD5(-) B-lymphocyte". Also, I assume the "mature NK-cell neoplasms" placed in the bottom right corner of the figure, would be more fitting in the top right corner instead, together with the depicted NK-cells.
Round 2
Reviewer 1 Report
Authors have satisfactorily addressed all my comments and suggestions.
Author Response
Thank you very much for your comments, critical review and insightful observations that have helped to improve the purpuse of the manuscript.